| Open Peer Review | Biotechnology | Methods and Protocols

# A benchmark of optimization solvers for genome-scale metabolic modeling of organisms and communities

Daniel Machado[1]

ABSTRACT   Genome-scale metabolic modeling is a powerful framework for predicting metabolic phenotypes of any organism with an annotated genome. For two decades, this framework has been used for the rational design of microbial cell factories. In the last decade, the range of applications has exploded, and new frontiers have emerged, including the study of the gut microbiome and its health implications and the role of microbial communities in global ecosystems. However, all the critical steps in this framework, from model construction to simulation, require the use of powerful linear optimization solvers, with the choice often relying on commercial solvers for their well-known computational efficiency. In this work, I benchmark a total of six solvers (two commercial and four open source) and measure their performance to solve linear and mixed-integer linear problems of increasing complexity. Although commercial solvers are still the fastest, at least two open-source solvers show comparable performance. These results show that genome-scale metabolic modeling does not need to be hindered by commercial licensing schemes and can become a truly open science framework for solving urgent societal challenges.

IMPORTANCE   Modeling the metabolism of organisms and communities allows for computational exploration of their metabolic capabilities and testing their response to genetic and environmental perturbations. This holds the potential to address multiple societal issues related to human health and the environment. One of the current limitations is the use of commercial optimization solvers with restrictive licenses for academic and non-academic use. This work compares the performance of several commercial and open-source solvers to solve some of the most complex problems in the field. Benchmarking results show that, although commercial solvers are indeed faster, some of the open-source options can also efficiently tackle the hardest problems, showing great promise for the development of open science applications.

KEYWORDS   genome-scale modeling, metabolism, optimization methods

Genome-scale metabolic modeling is a mathematical framework for predicting the metabolic phenotype of an organism based on genomic and environmental information (1). Some of the first models were developed for industrially relevant organisms like *Escherichia coli* (2) and *Saccharomyces cerevisiae* (3) and used to find optimal intervention strategies for rational strain design (4). Recently, automated reconstruction tools (5, 6) have enabled the creation of large model collections (7), accounting for virtually any organism that has been sequenced. This has expanded the frontiers of research using genome-scale models to address multiple societal issues, from the study of the human gut microbiome and its health implications (7) to global microbial ecosystems of the soils (8) and oceans (9).

While this research field exploded with new simulation methods and software tools (10), one fundamental obstacle remained. This family of the so-called constraint-based

Editor Ying Zhang, University of Rhode Island, Kingston, Rhode Island, USA

Address correspondence to Daniel Machado, daniel.machado@ntnu.no.

The author declares no conflict of interest.

See the funding table on p. 7.

analysis and reconstruction methods (COBRA) requires efficient linear optimization solvers that can handle thousands of variables and constraints. The choice has often fallen upon commercial optimization solvers like CPLEX (IBM) and GUROBI (Gurobi Optimization LLC) due to their computational efficiency. The GNU Linear Programming Kit (GLPK) has been used as an open-source alternative, but it offers slower computation times and no longer seems to be actively maintained. This bottleneck in the selection of optimization solvers creates obstacles for users, who become restricted by complex licensing schemes, hindering genome-scale modeling from becoming a truly open science framework.

In this work, I benchmark the three solvers mentioned above together with three recent open-source solvers that might be competitive alternatives to commercial solvers. The benchmark considers two main problem formulations: linear problems (LPs), which are commonly used to run flux balance analysis (FBA) simulations (11) and mixed-integer linear problems (MILPs), used to solve problems with any kind of selection constraints (implemented with binary or integer variables), such as optimal strain design (4) or gap-filling (12). Quadratic minimization problems have sometimes been used in the formulation of other objective functions (13) but were not addressed in this work, as they are not supported by all solvers (and can be reformulated as linear minimization problems, by replacing L2 with L1 norm).

## RESULTS

### LP benchmark

Benchmarking results for solving LPs (see Materials and Methods) are presented in Fig. 1. For single-species FBA, it can be observed that GUROBI was the fastest solver, followed by CPLEX, whereas SCIP was the slowest solver for models of small size. COIN starts with a performance comparable to HiGHS and GLPK, but it deteriorates for larger problem sizes, indicating some difficulty with scaling up. All problems are solved in a time scale of milliseconds, with negligible variation in runtime between solver runs. The largest model (Recon3D), with approximately 10K variables and 6K constraints, can be solved in a range of 0.1–1.0 seconds.

The results for the simulation of microbial communities are comparable to single-species FBA. GUROBI maintains its remarkable efficiency, and three open-source solvers (SCIP, HIGHs, and GLPK) perform very similarly, with a stable computational time that seems to approximate a complexity of $O(n \cdot \log(n))$. However, CPLEX suddenly drops in performance for communities with more than four members. The increase in computation time, by approximately 100-fold, is abrupt but stable. A closer inspection of the solver output logs reveals that this extra time is dedicated to internal post-processing after the optimal solution is found. COIN also decreases in performance for larger communities.

These two benchmarks (single species and community simulation) were repeated using parsimonious FBA (pFBA), which differs from standard FBA by an additional optimization step that minimizes the sum of the absolute values of all fluxes (Fig. S1). The calculation of the absolute values requires splitting the fluxes of reversible reactions into double the number of variables, considerably increasing the problem size. Indeed, it can be observed that computation time increases in a stable proportion relative to the FBA benchmark. The sudden drop in performance for CPLEX is now quite steep when switching from communities of three members to four. All solvers rank similarly in comparison to the previous benchmark, except for GLPK, which became the slowest solver for the largest problems.

To understand if the efficiency of the solvers could be related to their memory management, the benchmarks were monitored for maximum memory usage during execution (Fig. S2a). Each solver has a baseline memory usage of only a few hundred MB (circa 175 MB). For the commercial solvers, the memory usage barely increases, even for the largest problems. These low memory requirements are not surprising considering that, for instance, the stoichiometric matrix of the largest model (Recon3D) occupies

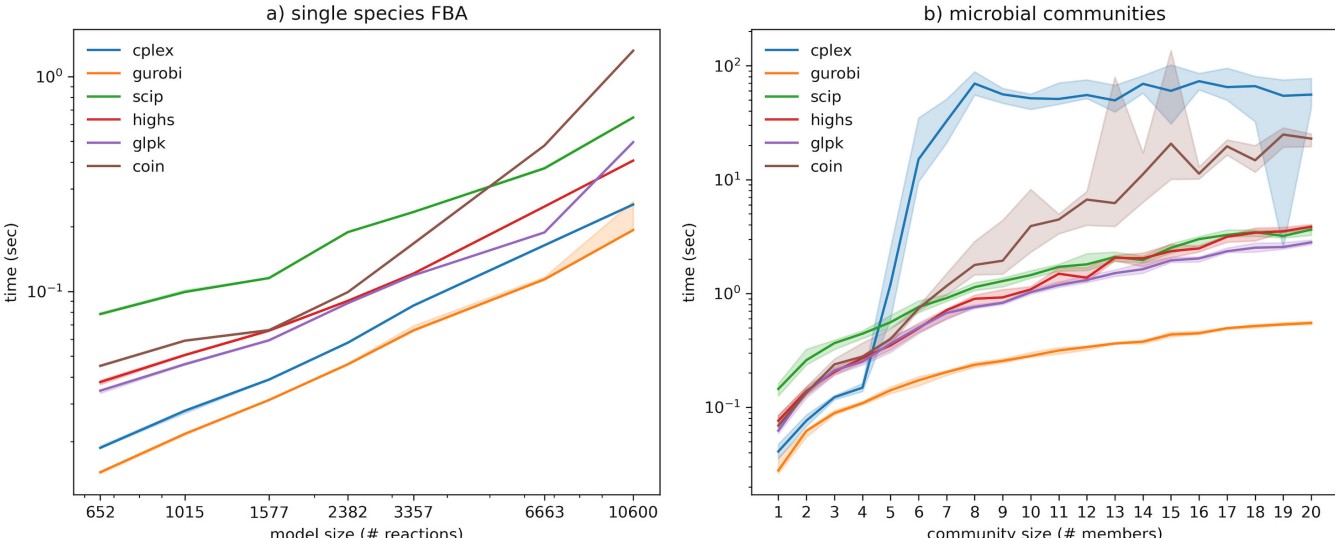

**FIG 1** Benchmarking results for LP formulations. (A) Single-species FBA. The horizontal axis (log scale) represents the number of continuous variables (number of reactions in each model). (B) Simulation of microbial communities. In both panels, the lines represent the median of 10 simulations and the error bands represent the interquartile range.

under 50 KB of memory. On the other hand, the open-source solvers do show a moderate memory increase. SCIP shows the highest increase, from 172 MB (smallest model) to 261 MB (largest model). Nonetheless, these are considerably low memory requirements for any modern computer and are unlikely to cause limitations in performance.

Finally, the impact of the solving strategy was also evaluated. LPs can be solved using the simplex algorithm (primal or dual formulation) or the barrier algorithm (also known as interior point method). According to their documentation, all the open-source solvers use an implementation of the simplex algorithm as their default strategy and allow the user to (optionally) select between primal simplex, dual simplex, or barrier (except for SCIP). GUROBI and CPLEX document using a multi-threaded strategy by default, running different algorithms in parallel until the fastest is finished. To better understand the influence of algorithm selection concerning execution time, the microbial community simulation benchmark was repeated for each algorithm option (Fig. 2). The results are quite surprising and show not only that different algorithms work better for different solvers but also that the default strategy is not always the best. Although barrier is the slowest method for both CPLEX and GUROBI, the default strategy seems to yield a similar computational cost. In fact, CPLEX no longer shows a drop in performance when one of the simplex methods is explicitly selected. SCIP only implements primal and dual simplex and yields the same results with both methods. HiGHS uses dual simplex by default but shows greater efficiency with barrier. GLPK is most efficient with its default method, primal simplex, and surprisingly better with barrier than dual simplex. COIN performs the best with primal simplex, although it uses dual simplex by default. Its performance with barrier is very unstable and the worst among all solvers. In conclusion, the LP solving strategy does have a significant impact and should be carefully selected for each solver.

## MILP benchmark

MILPs were formulated to determine minimal medium compositions for the single-species models (see Materials and Methods). As expected, MILPs have considerably higher runtimes than LPs, in the order of seconds to minutes (Fig. 3A). The variation between runtimes (in the order of milliseconds) is again negligible. GUROBI and CPLEX are once again the fastest solvers. In this case, COIN is clearly the slowest solver, with a runtime, on average, one order of magnitude above the others. For smaller problems, GLPK has a runtime comparable to SCIP and HiGHS, but its performance deteriorates for

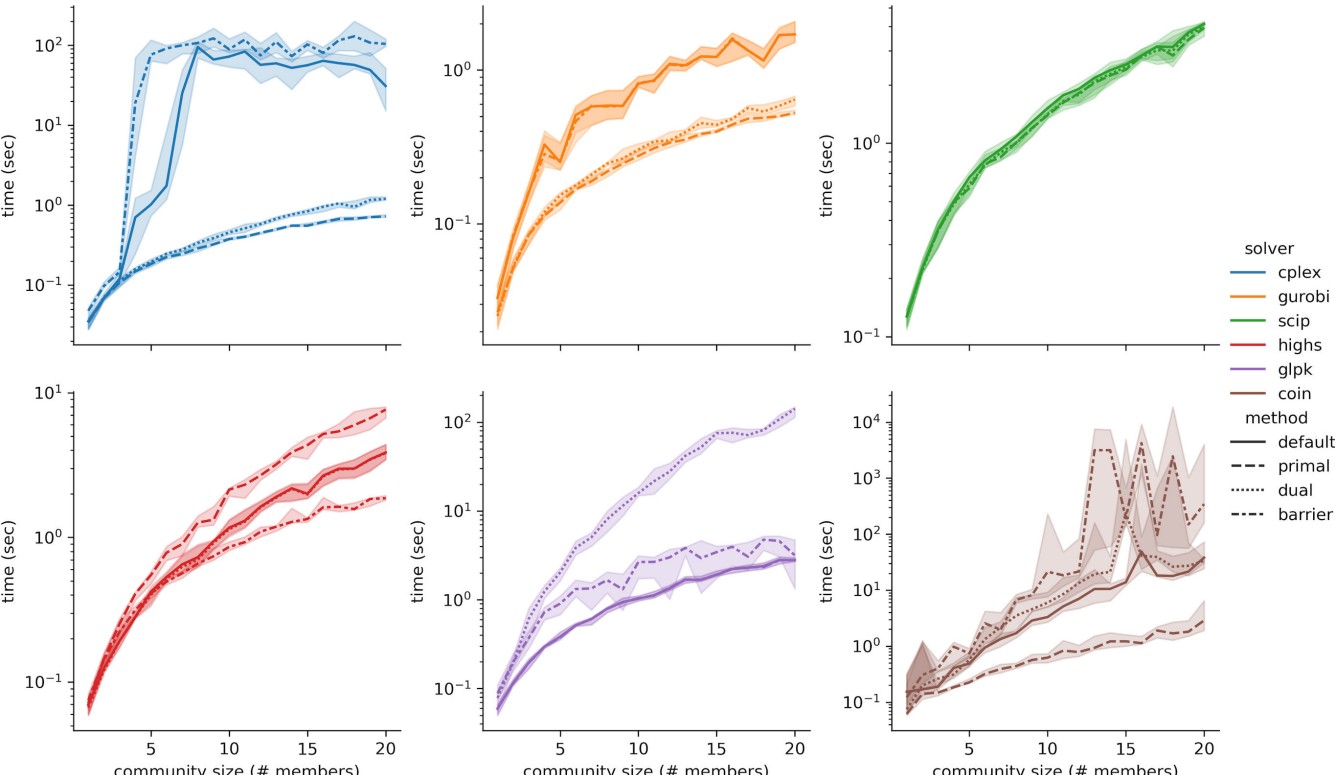

**FIG 2** Benchmarking results for different LP solving algorithms using microbial communities. Lines represent the median of 10 simulations and the error bands represent the interquartile range.

larger problems, indicating issues with scalability. Both GLPK and COIN failed to solve the largest problem (Recon3D) within 1 week of runtime.

Interestingly, the runtime for solving MILPs did not increase monotonously with the size of the models. For instance, the iAF1260 model (299 integer variables) took longer to solve than the larger iYS17120 model (472 integer variables), regardless of the choice of the solver. This indicates that other factors (such as model-specific constraints resulting from the network topology and stoichiometric coefficients) can contribute to the time required to find the optimal solution. The memory usage of the solvers was again compared (Fig. S2b). Unlike runtime, the memory requirements do not significantly increase in comparison to those of solving LPs. The commercial solvers again maintain a stable memory size regardless of the size of the model, whereas a stable increase is observed for the open-source solvers. SCIP reaches the highest, yet still modest, memory requirement of 410 MB when solving the Recon3D model. Hence, memory does not appear to be a critical factor for solving either LPs or MILPs.

To further analyze the sensitivity of the solvers toward perturbations in the MILP formulations, the Recon3D model was solved 1,000 times with random subsets of 312 binary variables (i.e., considering only 20% of all uptake reactions). The reduced problem complexity allowed to obtain performance results for GLPK and COIN (Fig. 3B). It can be observed that the solvers fall into three main categories: the fast solvers, CPLEX and GUROBI (both with an average runtime of 0.74 seconds); the intermediate solvers, SCIP and HiGHS (3.5 and 5.4 seconds); and the very slow solvers, GLPK and COIN (1.3 and 2.4 hours, respectively).

It is quite clear that the execution time can vary significantly for problems of the same size, especially for the slowest solvers (Fig. 3B). To solve an MILP, every solver implements some variation of the branch-and-cut algorithm. This involves solving an initial non-integer relaxation (LP) of the problem and then exploring alternative branches with feasible values for the integer variables. Each node to be evaluated in the tree requires

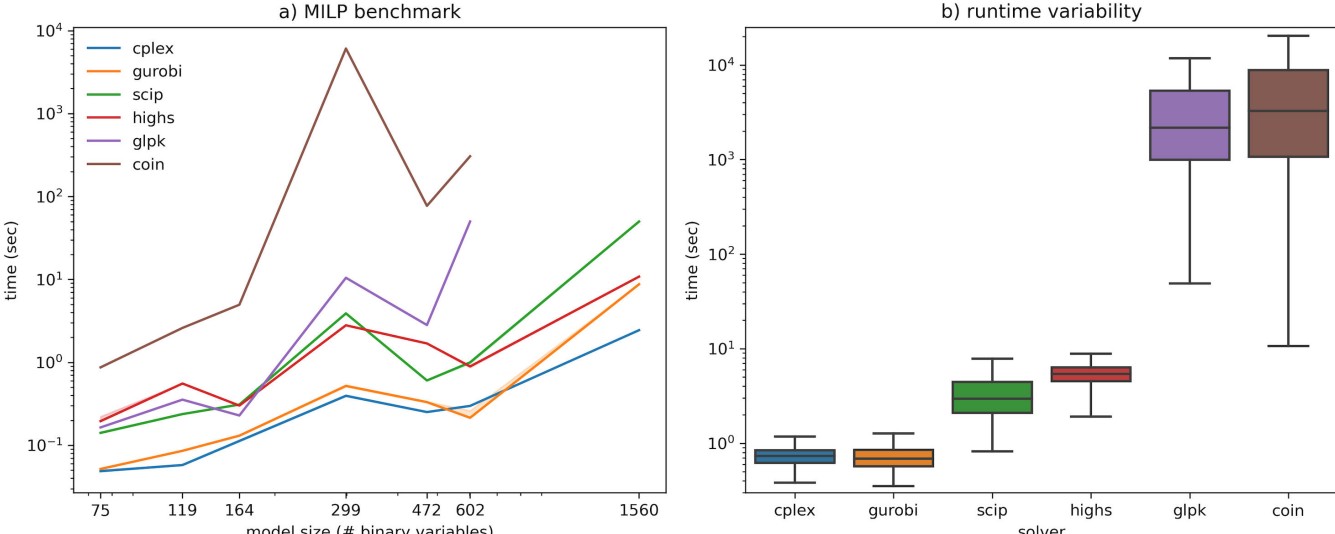

**FIG 3** Benchmarking results for MILP formulations. (A) Computation time for each model. The horizontal axis (log-scale) represents the number of binary variables (number of external metabolites in each model). Interquartile variation is represented by error bands but is too small to be visible. GLPK and COIN failed to solve the Recon3D model within 1 week of runtime. (B) Runtime variability for Recon3D with random subsets of variables (1,000 replicates).

one LP simulation. During execution, the tree can be pruned to remove branches that are guaranteed to be suboptimal, avoiding a combinatorial explosion of the search space. These branching and pruning strategies, as well as other problem transformation heuristics, are specific for each solver. Looking at their execution logs, it seems that the fastest solvers can apply efficient heuristics that find the optimal solution after exploring a small number of nodes, whereas the slowest solvers get trapped in an exhaustive exploration of nodes. Hence, for applications where a suboptimal solution is sufficient, the user can consider applying a time limit to stop the execution long before the global optimum is found.

## DISCUSSION

Overall, this work shows that, although CPLEX and GUROBI are the fastest solvers, some of the open-source solvers offer comparable performance. Even for the Recon3D model, with over 10K reactions, it is possible to run an FBA simulation in less than a second, and (except for GLPK and COIN) it is possible to solve an MILP with over 1.5K binary variables in just a few seconds. Among the open-source solvers, SCIP and HiGHS performed the best for the most demanding LP and MILP formulations. In addition, although SCIP had the highest memory requirements, a few hundred MBs are still modest requirements for any modern computer.

In this work, I opted to run all solvers with their default parameter configurations. Each solver can be potentially fine-tuned by adjusting internal parameters (tolerances, pre-solving, search strategies, etc.). Since each solver implements a different set of parameters, such fine-tuning is outside the scope of this work. However, modifying the default LP solving strategy did have a significant impact on the execution time of most solvers. Considering that optimization solvers are applied in several areas of research, their default settings are most likely a compromise between versatility and efficiency. Therefore, it is recommended that the selection of a solver is followed by a domain-specific adjustment of its parameters. Nonetheless, it was confirmed that all solvers returned the same objective values for every problem. The solution degeneracy typical of genome-scale models can, of course, result in alternative flux distributions with the same objective value. It was also observed that only the commercial solvers used multiple cores during execution. Parallel computation might thus be a key factor in improving the efficiency of open-source solvers.

Genome-scale metabolic modeling is a powerful framework to map genotypes to phenotypes and help to develop biotechnological solutions to address many of today's societal challenges. As we try to understand the impact of environmental changes in the planetary microbiome or the impact of our own microbes and diets in our health, more detailed models and sophisticated simulation tools will emerge. The open-source solvers used in this work are currently available for genome-scale modeling through a core simulation library (14) and can be easily interfaced with any other tool. This is a fundamental step for the ongoing integration of genome-scale modeling into complex bioinformatic workflows (15) that will help untangle the metabolic complexity of our biosphere.

## MATERIALS AND METHODS

### Software setup

Six optimization solvers (Table 1) were installed in a local machine (8-core CPU with 16 GB of RAM). The open-source solvers were installed using the conda package manager. For the commercial solvers, software installers and free academic licenses were obtained from the respective websites. All solvers were executed with their default parameter configurations. The PuLP library (16) (version 2.7.0) was used as a common interface to interact with all the solvers. All simulations were performed using the ReFramed library (14) (version 1.3.0).

### LP benchmark

The first LP benchmark consisted of single-species FBA simulations with growth maximization (equation 1). Seven genome-scale metabolic models (Table 2) were downloaded from the BiGG Models (17) database. These models were selected to represent a wide variety of model sizes (uniformly distributed on a logarithmic scale; see Fig. 1). The growth conditions for each organism were defined as complete media (with a maximum uptake rate of 1 mmol/gDW/h for every extracellular compound in the models). The simulations were repeated 10 times for each combination of model and solver to better estimate the computational time.

$$\max v_{growth}$$
$$\text{s.t.}$$
$$S \cdot v = 0$$
$$lb \leq v \leq ub$$

(1)

A second benchmark was performed for the simulation of microbial communities with up to 20 members. The models were downloaded from the EMBL GEMs collection (6) of 5,587 bacterial models. For each community size, 10 different community models were created by random sampling of species from the model collection. The communities were assembled using a compartmentalized approach and simulated by maximizing the sum of the growth rates of all members.

**TABLE 1** List of optimization solvers used in this work

| Solver | Version | Release date | License |
|---|---|---|---|
| CPLEX | 22.1.0 | March 2022 | Commercial[a] |
| GUROBI | 10.0.1 | February 2023 | Commercial[a] |
| GLPK | 5.0 | December 2020 | GPL v3 |
| COIN (CLP/CBC) | 2.10.8 | May 2022 | EPL 2.0 |
| SCIP | 8.0.3 | December 2022 | Apache 2.0 |
| HiGHS | 1.5.0 | February 2023 | MIT |

[a]The commercial solvers are free for academic use after obtaining the respective licenses.

**TABLE 2**  List of genome-scale metabolic models used in this work

| Model | Organism | Reactions | Metabolites | External |
|---|---|---|---|---|
| iLJ478 | *Thermotoga maritima* | 652 | 720 | 75 |
| iCN718 | *Acinetobacter baumannii* | 1,015 | 888 | 119 |
| iMM904 | *Saccharomyces cerevisiae* | 1,557 | 1,226 | 164 |
| iAF1260 | *Escherichia coli* | 2,382 | 1,668 | 299 |
| iYS1720 | *pan-Salmonella* | 3,357 | 2,436 | 472 |
| iCHOv1 | *Cricetulus griseus* | 6,663 | 4,456 | 602 |
| Recon3D | *Homo sapiens* | 10,600 | 5,835 | 1,560 |

**MILP benchmark**

The MILP benchmark consisted of finding a minimal growth medium for each model in Table 2. The problem is implemented by minimizing the total number of active uptake reactions in each model subject to a non-zero growth rate (equation 2). The number of binary variables corresponds to the number of external metabolites in each model. The computation was repeated 10 times for every combination of model and solver.

$$\min \sum_{i=1}^{n} y_i \quad \forall i \in R_{\text{exchange}}$$
$$\text{s.t.}$$
$$S \cdot v = 0$$
$$lb \leq v \leq ub \tag{2}$$
$$v_{\text{growth}} \geq 1$$
$$v_i \geq -100\, y_i$$
$$y_i \in \{0, 1\}$$

An additional robustness test was performed for solving MILPs with the Recon3D model using only a random sample of 312 external metabolites (i.e., 20% of the original number of binary variables). The test was performed 1,000 times for each solver. In this case, due to their long computation times, GLPK and COIN were executed in a computing cluster with a configuration similar to the local environment (8 CPU cores with a clock speed of 3.2 GHz and 16 GB of RAM).

## ACKNOWLEDGMENTS

This work was supported by ELIXIR Norway (funded by the Research Council of Norway).

## AUTHOR AFFILIATION

[1]Department of Biotechnology and Food Science, Norwegian University of Science and Technology (NTNU), Trondheim, Norway

## AUTHOR ORCIDs

Daniel Machado  http://orcid.org/0000-0002-2063-5383

## FUNDING

| Funder | Grant(s) | Author(s) |
|---|---|---|
| Norges Forskningsråd (Forskningsrådet) | 322392 | Daniel Machado |

## AUTHOR CONTRIBUTIONS

Daniel Machado, Conceptualization, Data curation, Formal analysis, Investigation, Methodology, Software, Validation, Writing – original draft, Writing – review and editing

## DATA AVAILABILITY

Source code and data are openly available at GitHub.

## ADDITIONAL FILES

The following material is available online.

### Supplemental Material

**Fig. S1 (mSystems00833-23-S0001.docx).** Benchmark using pFBA simulation.
**Fig. S2 (mSystems00833-23-S0002.docx).** Memory benchmark.

### Open Peer Review

**PEER REVIEW HISTORY (review-history.pdf).** An accounting of the reviewer comments and feedback.

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
