## [Reviewer comments · mSystems]

A benchmark of optimization solvers for genome-scale metabolic modeling of organisms and communities

Daniel Machado

Corresponding Author(s): Daniel Machado, Norges teknisk-naturvitenskapelige universitet

Review Timeline:

Submission Date:	August 8, 2023
Editorial Decision:	September 27, 2023
Revision Received:	November 13, 2023
Accepted:	December 11, 2023

Editor: Ying Zhang

Reviewer(s): The reviewers have opted to remain anonymous.

Transaction Report:

DOI: <https://doi.org/10.1128/msystems.00833-23>

September 27, 2023

Prof. Daniel Machado
Norges teknisk-naturvitenskapelige universitet
Department of Biotechnology and Food Science
Trondheim
Norway

Re: mSystems00833-23 (A benchmark of optimization solvers for genome-scale metabolic modeling of organisms and communities)

Dear Prof. Daniel Machado:

Thank you for submitting your manuscript to mSystems. We have completed our review and I am pleased to inform you that, in principle, we expect to accept it for publication in mSystems. However, acceptance will not be final until you have adequately addressed the reviewer comments.

Specifically, please report memory/core usage and consider including comparisons with parsimonious FBA.

Upon resubmission, please provide point-by-point responses to each comment made by the reviewers, identifying the location of each revision by page and line number, in a file uploaded as "Response to Reviewers." Additionally, please include a copy of the revised manuscript in which all changes are identified in a file uploaded as "Marked Up Manuscript." In the revised manuscript, please also include an "Importance" section following "Abstract" to highlight the importance and main contribution of this work.

Preparing Revision Guidelines

Please return the manuscript within 60 days; if you cannot complete the modification within this time period, please contact me. If you do not wish to modify the manuscript and prefer to submit it to another journal, please notify me of your decision immediately so that the manuscript may be formally withdrawn from consideration by mSystems.

The ASM Journals program strives for constant improvement in our submission and publication process. Please tell us how we

can improve your experience by taking this quick Author Survey.

Sincerely,

Ying Zhang

Editor, mSystems

Journals Department
Reviewer comments:

Reviewer #1 (Comments for the Author):

mSystems: Machado

In this work, the author performed a comparative analysis of computational solver packages for linear programming (LP) problems with a particular focus on Flux Balance Analysis (FBA) of COBRA (constrained-based analysis and reconstruction) models. In total, six solvers were included in the comparison: two commercial packages-CPLEX and GUROBI and four non-commercial packages-GLPK, COIN, SCIP, and HiGHS. Two types of FBA were considered, the standard LP and a mixed integer LP (MILP). The results conclusively showed that the commercial packages, particularly GUROBI, are the most efficacious solvers for these types of problems. Among the non-commercial packages, HiGHS has a notable performance (efficacy and robustness), but still is slower than the commercial packages.

As a practitioner in the field, I found the work to be of high quality and well written despite the unsurprising results. Still, the comparison and findings will be useful, especially the option of free, albeit slower, non-commercial packages.

Major Comments:

1. I am wondering how much the efficacy of commercial packages is due to the optimization algorithms used for solving LP and MILP problems. I think the value of this work can improve markedly given information and comparison on the specific optimization methods in different packages.
2. The quality of the results, in addition of the computational efficacy, should be presented. In the Discussion mentioned that different packages produced the same optimal values. This result should be provided in the Result.
3. Besides computational speed, I wonder if memory requirement ever become a (limiting) factor when scaling up model size. For example, I wonder if the unexpected trends observed for COIN and CPLEX have to do with the memory requirement and/or management.
4. The author mentioned that a computer with multiple cores were used. Do all packages take advantage of these multiple cores?
5. If I understand correctly, the FBA considered here is the standard formulation. Providing the optimization problems explicitly in the Methods will give clarity.
6. Last but not least, the omission of parsimonious FBA from the comparison is rather surprising, especially considering the author's 2014 PLoS Comp. Biology paper.

Reviewer #2 (Comments for the Author):

This thorough and well executed assessment of numerical solvers used in the context of constraint-based modeling is timely and extremely useful for the practitioner. I only have minor suggestions for improvement. One point considers the growth media used in the simulations, these should shortly be described.

In the absence of line numbers, I quote original text fragments prior to my remaining comments:

1. "creation of large model collections": consider referencing additionally the AGORA / AGORA2 model collections, as the human gut microbiome is mentioned multiple times as an area of application
2. "to address most societal issues": this appears to be quite an exaggeration, consider a more modest claim
3. "reconstruction (COBRA) methods": consider moving the parenthesis after "methods" to improve reading flow
4. "requires efficient linear optimization problems": consider replacing "problems" by "solvers"
5. "due to their known computational efficiency": consider dropping "known"
6. "that can be a promising replacement for": consider replacing by "that might be competitive alternatives to"

7. "and other kinds of theoretical yield analysis": explain in detail; FBA mentioned in the beginning of the sentence could equally be used to optimize for growth, yields (or other objectives).
8. "Quadratic minimization problems": elaborate on their application, as has been done before for LP & MILP
9. Table 1: consider adding the availability of free academic licenses for CPLEX & GUROBI
10. "of single-species FBA": consider adding "a" after of
11. "(uniformly distributed on a ...)": consider adding a reference to the Figure, which makes this statement easier to understand
12. Table 2: consider replacing "External" by "External Metabolites" or "External Compounds" for clarity
13. "for simulation of microbial communities": consider adding a "a" after "for"
14. "A second benchmark ...": clarify whether each random community was run 10 times as was done in the other benchmarks, or if 10 different random communities were simulated one time for each diversity level; the latter option would explain the larger runtime variation in Figure 1 b) and should be discussed in the text accordingly.
15. "(8 CPU,)" consider mentioning additionally clock speed
16. "with a negligible": consider dropping "a"
17. "replicates": consider to replace by "simulation/solver runs", as not a biological/technical replicate is meant, but repetitions are only used for obtaining better runtime estimates
18. Figures: consider adding a label to x-axis (model size), not only describing x-axis in legend
19. "4-5": consider replacing by "4 to 5"
20. "This change", consider by replacing by "This increase"
21. " $O(n \log(n))$ ": replace "*" by proper multiplication sign, also: could this be added as a line in the figure for reference? Are similar terms derivable for the other benchmarks?
22. "again becomes negligible": consider replacing by "is again negligible"
23. "failed to solve": consider to more correctly include the next sentence: "failed to solve ... within one week of runtime", also correct this statement in the legend to Figure 2.
24. Figure 2b): the text refers to "robustness" and "sensitivity analysis" is used as a term in the figure, which might be misleading. Consider using "runtime variability" (or similar) to avoid confusion.
25. "too small to be detected": consider replacing by "too small to be visible"
26. "are still the fastest": consider dropping "still"
27. "MILP with", consider replacing by "MILP problem with"
28. "the best": consider dropping "the"
29. "sudden lack of performance ...": consider replacing by "sudden drop in performance of CPLEX for large microbial communities"
30. "modeling, through a core simulation library": drop comma

Reviewer #1 (Comments for the Author):

In this work, the author performed a comparative analysis of computational solver packages for linear programming (LP) problems with a particular focus on Flux Balance Analysis (FBA) of COBRA (constrained-based analysis and reconstruction) models. In total, six solvers were included in the comparison: two commercial packages-CPLEX and GUROBI and four non-commercial packages-GLPK, COIN, SCIP, and HiGHS. Two types of FBA were considered, the standard LP and a mixed integer LP (MILP). The results conclusively showed that the commercial packages, particularly GUROBI, are the most efficacious solvers for these types of problems. Among the non-commercial packages, HiGHS has a notable performance (efficacy and robustness), but still is slower than the commercial packages.

As a practitioner in the field, I found the work to be of high quality and well written despite the unsurprising results. Still, the comparison and findings will be useful, especially the option of free, albeit slower, non-commercial packages.

I thank the reviewer for the positive feedback and for all the constructive criticism. Please find below a response to the comments.

Major Comments:

1. I am wondering how much the efficacy of commercial packages is due to the optimization algorithms used for solving LP and MILP problems. I think the value of this work can improve markedly given information and comparison on the specific optimization methods in different packages.

For solving LPs all solvers (except SCIP) offer the option to alternate between simplex (primal or dual) and the barrier algorithm (also known as the interior point method). I performed an additional benchmark to compare the different algorithms for each solver and expanded the results and discussion. Regarding MILPs the scenario is a bit more complex. Each solver implements some variation of the branch-and-cut method with its own additional heuristics. These heuristics are not described in detail, especially for the commercial solvers since they are part of their intellectual property for competitive advantage.

2. The quality of the results, in addition of the computational efficacy, should be presented. In the Discussion mentioned that different packages produced the same optimal values. This result should be provided in the Result.

I considered plotting this, but since the results are effectively the same across all solvers and replicated simulations, the result would be a single overlapping dot, which is not very informative. Nonetheless, all the simulation results are stored as CSV tables in the supplementary github repository, as well as the code that generated the results. Any user can therefore confirm and reproduce this observation.

3. Besides computational speed, I wonder if memory requirement ever become a (limiting) factor when scaling up model size. For example, I wonder if the unexpected trends observed for COIN and CPLEX have to do with the memory requirement and/or management.

Thank you for the suggestion. I have now also performed memory benchmarks (supp fig 2). It seems that memory is not an issue. Most executions required a maximum of 200 – 300 MB of memory, even for the largest models. These observations have been added to the results section.

4. The author mentioned that a computer with multiple cores were used. Do all packages take advantage of these multiple cores?

It seems that only the commercial solvers use multiple cores. This point has been added to the discussion.

5. If I understand correctly, the FBA considered here is the standard formulation. Providing the optimization problems explicitly in the Methods will give clarity.

The equations for the standard FBA formulation (LP problem) and for computing a minimal medium (MILP problem) are now provided in the methods section.

6. Last but not least, the omission of parsimonious FBA from the comparison is rather surprising, especially considering the author's 2014 PLoS Comp. Biology paper.

I had initially also performed pFBA simulations. I later removed them because the results weren't much more informative than the FBA simulations alone. Nevertheless, I agree that having a comparison with a second method might be valuable. I have re-included the pFBA simulations in the paper (supp fig. 1).

Reviewer #2 (Comments for the Author):

This thorough and well executed assessment of numerical solvers used in the context of constraint-based modeling is timely and extremely useful for the practitioner. I only have minor suggestions for improvement. One point considers the growth media used in the simulations, these should shortly be described.

I thank the reviewer for the very positive comments and for the detailed proof-reading that has considerably improved the quality of the text. I have made all requested modifications accordingly. I think that a point-by-point response is not really appropriate in this case.

In the absence of line numbers, I quote original text fragments prior to my remaining comments:

1. "creation of large model collections": consider referencing additionally the AGORA / AGORA2 model collections, as the human gut microbiome is mentioned multiple times as an area of application
2. "to address most societal issues": this appears to be quite an exaggeration, consider a more modest claim
3. "reconstruction (COBRA) methods": consider moving the parenthesis after "methods" to improve reading flow
4. "requires efficient linear optimization problems": consider replacing "problems" by "solvers"
5. "due to their known computational efficiency": consider dropping "known"
6. "that can be a promising replacement for": consider replacing by "that might be competitive alternatives to"
7. "and other kinds of theoretical yield analysis": explain in detail; FBA mentioned in the beginning of the sentence could equally be used to optimize for growth, yields (or other objectives).
8. "Quadratic minimization problems": elaborate on their application, as has been done before for LP & MILP
9. Table 1: consider adding the availability of free academic licenses for CPLEX & GUROBI
10. "of single-species FBA": consider adding "a" after of
11. "(uniformly distributed on a ...)": consider adding a reference to the Figure, which makes this statement easier to understand
12. Table 2: consider replacing "External" by "External Metabolites" or "External Compounds" for clarity
13. "for simulation of microbial communities": consider adding a "a" after "for"
14. "A second benchmark ...": clarify whether each random community was run 10 times as was done in the other benchmarks, or if 10 different random communities were simulated one time for each diversity level; the latter option would explain the larger runtime variation in Figure 1 b) and should be discussed in the text accordingly.

15. "(8 CPU,)" consider mentioning additionally clock speed
16. "with a negligible": consider dropping "a"
17. "replicates": consider to replace by "simulation/solver runs", as not a biological/technical replicate is meant, but repetitions are only used for obtaining better runtime estimates
18. Figures: consider adding a label to x-axis (model size), not only describing x-axis in legend
19. "4-5": consider replacing by "4 to 5"
20. "This change", consider by replacing by "This increase"
21. " $O(n \cdot \log(n))$ ": *replace "*" by proper multiplication sign*, also: could this be added as a line in the figure for reference? Are similar terms derivable for the other benchmarks?
22. "again becomes negligible": consider replacing by "is again negligible"
23. "failed to solve": consider to more correctly include the next sentence: "failed to solve ... within one week of runtime", also correct this statement in the legend to Figure 2.
24. Figure 2b): the text refers to "robustness" and "sensitivity analysis" is used as a term in the figure, which might be misleading. Consider using "runtime variability" (or similar) to avoid confusion.
25. "too small to be detected": consider replacing by "too small to be visible"
26. "are still the fastest": consider dropping "still"
27. "MILP with", consider replacing by "MILP problem with"
28. "the best": consider dropping "the"
29. "sudden lack of performance ...": consider replacing by "sudden drop in performance of CPLEX for large microbial communities"
30. "modeling, through a core simulation library": drop comma

Re: mSystems00833-23R1 (A benchmark of optimization solvers for genome-scale metabolic modeling of organisms and communities)

Dear Prof. Daniel Machado:

Your manuscript has been accepted, and I am forwarding it to the ASM production staff for publication. Your paper will first be checked to make sure all elements meet the technical requirements. ASM staff will contact you if anything needs to be revised before copyediting and production can begin. Otherwise, you will be notified when your proofs are ready to be viewed.

Featured Image Submissions: If you would like to submit a potential Featured Image, please email a file and a short legend to mSystems@asmusa.org. Please note that we can only consider images that (i) the authors created or own and (ii) have not been previously published. By submitting, you agree that the image can be used under the same terms as the published article. File requirements: square dimensions (4" x 4"), 300 dpi resolution, RGB colorspace, TIF file format.

Sincerely,
Ying Zhang
Editor
mSystems

Reviewer #1 (Comments for the Author):

Thank you for addressing my comments thoroughly.

The additional information about the use of multiple cores by commercial software package but not by the free packages, is important and would be useful for those who use the free optimization tool.

Reviewer #2 (Comments for the Author):

Excellent, I have no further comments.